# Comparative Study on the Effectiveness of Three Inoculation Methods for *Valsa sordida* in *Populus alba* var. *pyramidalis*

**DOI:** 10.3390/biology13040251

**Published:** 2024-04-09

**Authors:** Wanna Shen, Long Pan, Yuchen Fu, Yutian Suo, Yinan Zhang, Huixiang Liu, Xiaohua Su, Jiaping Zhao

**Affiliations:** 1State Key Laboratory of Tree Genetics and Breeding, Institute of Ecological Conservation and Restoration, Chinese Academy of Forestry, Beijing 100091, China; 2Shandong Research Center for Forestry Harmful Biological Control Engineering and Technology, College of Plant Protection, Shandong Agricultural University, Taian 271002, China; 3State Key Laboratory of Tree Genetics and Breeding, Research Institute of Forestry, Chinese Academy of Forestry, Beijing 100091, China

**Keywords:** punching inoculation, burning inoculation, toothpick inoculation, inoculation position

## Abstract

**Simple Summary:**

Pathogens of diseases in branches and trunks are commonly inoculated through punching, burning, and toothpick inoculation. At present, there is a lack of comparative analyses of the inoculation outcomes of these three methods. We determined that toothpick inoculation is an optimal option for studying the pathogenesis of *Valsa sordida* in six-year-old *Populus alba* var. *pyramidalis*, providing technical support for research on poplar diseases and offering a theoretical basis for the inoculation of other diseases in branches and trunks.

**Abstract:**

A key step in the study of tree pathology is the identification of an appropriate method for inoculating pathogens of diseases in branches and trunks. Pathogens of diseases in branches and trunks are commonly inoculated through punching, burning, and toothpick inoculation. However, there is a lack of comparative analyses of the inoculation outcomes of these three methods. In this work, six-year-old *P. alba* var. *pyramidalis* were inoculated with *V. sordida* using punching, burning, and toothpick techniques to investigate the differences in the effectiveness of these inoculation methods. Results reveal that the incidence rate was 93.55% in the toothpick inoculation group, significantly higher than the 80.65% in the burning inoculation group (chi-square, *n* = 90, *p* = 0.007), while punching inoculation exhibited significant pathological responses in the early stages, with spontaneous healing in the later stage. Additionally, toothpick inoculation was more efficient in inducing *Valsa* canker when inoculating the pathogen at the bottom of the tree, with lower intra- and inter-row spacing (stand density) providing better outcomes than higher intra- and inter-row spacing. The results of this study demonstrate that toothpick inoculation is an optimal option for studying the artificial inoculation of *V. sordida* in six-year-old *P. alba* var. *pyramidalis*, providing technical support for research on poplar diseases and offering a theoretical basis for the inoculation of other diseases in the branch and trunk.

## 1. Introduction

Plant pathogen inoculation plays a crucial role in plant disease research. Early inoculation experiments aim primarily at determining the pathogenicity of a given pathogen and establishing its role in causing disease. In later stages, such experiments have been widely employed in the research of pathology, such as the infection process, pathogenicity, genetic differentiation, the differentiation and genetics of plant resistance to pathogens, and resistance breeding. There is a diverse range of plant pathogens, each requiring specific inoculation methods, with a greater extent of disease being the criterion for choosing the optimal inoculation method [1]. The extent of disease in plant populations can be quantified by the incidence rate, the degree of disease severity, and the disease severity index. Particularly, the disease severity index provides a comprehensive assessment of the incidence rate and the degree of disease severity [2]. Previous research has indicated that the choice of a suitable inoculation method, timing, and site has a significant impact on the inoculation outcomes [3,4].

Stem diseases occur in the channel tissues (trunk, and small and large branches) connecting tree roots and leaves, causing xylem and phloem tissue lesions and necrosis, resulting in tree wilt [5]. A range of distinctive techniques have been developed for the inoculation of pathogens which can induce stem diseases during the study process. For instance, *Botryosphaeria dothidea* and *Valsa sordida*, which can induce stem canker and canopy dieback, were inoculated in the branches of *Populus beijingensis* clones via punching to study the molecular mechanisms of plant responses to pathogen infectionand the study results showed a significant up-regulation of the majority of disease-resistant genes in poplar [6]. *V. ceratosperma* was inoculated in isolated mature branches and twigs of apple trees using the burning technique, burning the bark of branches or trunks with a heated iron to make a hole for the inoculation mycelium plug, and the branches exhibited higher resistance to the pathogen compared to the twigs, suggesting a positive correlation between branch resistance and tree age [7]. By inoculating *Dothiorella gregaria* (the once used anamorph synonym of *B. dothidea*) in two-year-old *P. simonii Carr × nigra* var. *italica* using intact skin, needles, wounded skin, and burning techniques, the study results found that burning inoculation was associated with a higher incidence of diseases in the branches and trunk and faster lesion expansion [8]. *Diaporthe* spp. was inoculated in soybeans, using the toothpick and trunk wounding techniques, noted that the length of lesions induced by toothpick inoculation and the disease severity index significantly surpassed those in the trunk wounding inoculation group [9]. Collectively, punching, burning, and toothpick inoculation are common techniques for the inoculation of diseases in branches and trunks. However, the results of many studies using one-year-old seedlings or isolated trunks for inoculation may fail to represent the disease resistance of perennial plants.

*Populus alba* var. *pyramidalis* is an economically important tree and serves as a model plant in tree biology research [10]. It is widely distributed in the deserts and arid regions of northwest China. *P. alba* var. *pyramidalis* has been extensively used for timber production and ecological conservation, exhibiting significant economic value, which are attributed to its characteristics of fast growth, high biomass yield, and remarkable tolerance to stressors such as drought and salt [11,12]. Therefore, *P. alba* var. *pyramidalis* was selected as the object of inoculation in our experiment. *V. sordida* is a significant plant pathogen that can cause *Cytospora* canker (also known as *Valsa* canker) involving the branches and trunks of trees. This pathogen is widely distributed in the northern regions of China [13,14], posing a severe threat to poplars. In fast-growing poplar plantations, the incidence of *Valsa* canker ranges from 20% to 40%, reaching over 90% in some cases [15,16,17], causing substantial economic losses to commercial forestry. Against this background, *V. sordida* was investigated in this experiment.

Punching and burning techniques are most frequently used in studies on the pathogenicity and inoculation of *V. sordida* [6,18]. In contrast, the aforementioned toothpick inoculation technique has not been utilized in inoculation experiments with *V. sordida*, and there is a lack of systematic comparisons of these three inoculation methods for *P. alba* var. *pyramidalis*. In this study, punching, burning, and toothpick inoculation were employed to wound six-year-old *P. alba* var. *pyramidalis* to compare the inoculation outcomes of these methods by analyzing the incidence of *Valsa* canker and disease extent caused by *V. sordida* infection. Additionally, this study also elaborated the effects of intra- and inter-row spacing (stand density), inoculation position, and plant age, factors closely associated with the occurrence and development of diseases, on the inoculation outcomes, aiming to identify the optimal inoculation method and determine the ideal inoculation position as well as intra- and inter-row spacing when using this optimal method. The present study is expected to provide technical support for artificial inoculation research on poplar pathogens and serve as a theoretical basis for the inoculation of other diseases in branches and trunks.

## 2. Materials and Methods

### 2.1. Plants and Fungi

The objects of inoculation were nine six-year-old *P. alba* var. *pyramidalis* with a diameter at breast height of 18.18 ± 1.13 cm, exhibiting robust growth and free from plant diseases and insect pests, with intra- and inter-row spacing of 0.8 m × 0.8 m in six plants and 1.6 m × 1.6 m in the remaining trees. Additionally, 45 two-year-old *P. alba* var. *pyramidalis* with a ground diameter of 13.59 ± 1.31 mm, robust growth, and absence of diseases and pests were included. The six-year-old *P. alba* var. *pyramidalis* were cultivated at the experimental site of the Plant Physiology Laboratory, Institute of Ecological Conservation and Restoration, Chinese Academy of Forestry. The two-year-old *P. alba* var. *pyramidalis* cuttings were cultivated in plastic pots containing a mixed substrate (peat soil/perlite = 6:1) and grown at the same experimental site. In this study, the *V. sordida* strain CZC (NCBI accession numbers: MK994101 for rRNA-ITS and MN025273 for EF1αgene) and the *B. dothidea* strain CZA (NCBI accession numbers: MK990559 for rRNA-ITS and MN025271 for EF1αgene) were used as the primary fungal materials, respectively [19]. After activation, these strains were cultured on potato dextrose agar (PDA) at pH 6.0 and incubated in the dark at 28 °C for 7 days.

The six- and two-year-old *P. alba* var. *pyramidalis* were cultivated in the same experimental environment, and the fungi were activated from the identical strains. Therefore, the disease resistance of poplars and the pathogenicity of the fungi remained consistent throughout the study.

### 2.2. Punching, Burning, and Toothpick Inoculation Experiments on Six-Year-Old P. alba var. pyramidalis

#### 2.2.1. Punching Inoculation

One day before inoculation, the poplar trunk was cleaned with water, and the bark surface was sterilized with 75% alcohol. The trunk was air-dried before the inoculation experiment. The trunk was divided into the top and the bottom, with the upper located 80 cm above the ground and the lower 10 cm above the ground, leaving a 10 cm blank in the middle. Using a sterilized 5 mm puncher, circle-shaped holes (the depth of each hole was about 1 mm) were made in the top and the bottom of the trunk with a spacing of approximately 5 cm between holes. Each area had 15 inoculation holes. Three poplars were inoculated in total, including two with intra- and inter-row spacing of 0.8 m × 0.8 m and one with intra- and inter-row spacing of 1.6 m × 1.6 m. After punching, the periderm and phloem were removed for inoculation, and a 5 mm fungal clump was placed in the hole using a sterile inoculating needle. The inoculation sites were covered with plastic wrap for moisture retention and prevention of contamination. Simultaneously, holes were made 5 cm beside the inoculation sites of *V. sordida* to inoculate agar blocks as controls, with the top and the bottom containing five holes, respectively, to a total of ten holes as the controls in every poplar. The plastic wrap was removed at 1 week post inoculation. The lesion development at the inoculation sites was observed at days 10, 20, 30, 40, and 50 post inoculation. Disease markers were measured to analyze the impact of different inoculation sites as well as intra- and inter-row spacing on the development of *V. sordida* infection.

#### 2.2.2. Burning Inoculation

Similarly, the inoculation experiment was started after cleaning the poplar trunk with water, sterilization of the bark surface with 75% alcohol, and air-drying the trunk one day before inoculation. Using a heated 5 mm iron nail, the trunk was scorched until the bark turned brown. After cooling, the trunk was immediately inoculated with *V. sordida* clumps. Fifteen burnt sites were inoculated on the top and the bottom of the trunk, while five additional agar blocks were inoculated, respectively, in the top and the bottom as controls. The inoculation sites were covered with plastic wrap for moisture retention. Three poplars were inoculated in total, including two with intra- and inter-row spacing of 0.8 m × 0.8 m and one with intra- and inter-row spacing of 1.6 m × 1.6 m. At 1 week post inoculation, the plastic wrap was removed. Lesion development at the inoculation sites was observed at days 10, 20, 30, 40, and 50 post inoculation. Disease markers were measured to analyze the impact of different inoculation sites and intra- and inter-row spacing on the development of *V. sordida* infection.

#### 2.2.3. Toothpick Inoculation

Similarly, the inoculation experiment was initiated after cleaning the poplar trunk with water, sterilization of the bark surface with 75% alcohol, and air-drying the trunk one day before inoculation. A total of 15–20 sterilized toothpicks (length: 6.5 cm; diameter: 0.2 cm; Guanghui, Harbin, China) were arranged in parallel on a 9 cm PDA solid plate. CZC was inoculated onto the toothpicks, and the plate was incubated in the dark at 25 °C for 7 days until the toothpicks were completely covered by fungal mycelia. Before inoculation, holes were drilled 0.8–1.0 cm vertically into the trunk surface using a 0.15 cm drill bit. Following that, the fungus-inoculated toothpicks were quickly inserted into the holes. Fifteen fungus-inoculated toothpicks were placed in the top and the bottom of the trunk, respectively, while five additional sterile toothpicks were inserted as controls in the top and the bottom, respectively. The inoculation sites were covered with plastic wrap for moisture retention. Three poplars were inoculated in total, including two with intra- and inter-row spacing of 0.8 m × 0.8 m and one with intra- and inter-row spacing of 1.6 m × 1.6 m. At 1 week post inoculation, the plastic wrap was removed. The lesion development at the inoculation sites was observed at days 10, 20, 30, 40, and 50 post inoculation. Disease markers were measured to analyze the impact of different inoculation sites and intra- and inter-row spacing on the development of *V. sordida* infection.

### 2.3. Burning Inoculation Experiment on Two-Year-Old P. alba var. pyramidalis

Burning inoculation was employed to inoculate 15 two-year-old *P. alba* var. *pyramidalis* with the pathogenic fungi CZA and CZC, each containing 8 inoculation sites, with agar blocks serving as the controls. The development of lesions was observed, and disease markers were measured at 50 days post inoculation. The statistical analysis and data processing for these marker measurements were consistent with those applied in the inoculation experiment on the six-year-old *P. alba* var. *pyramidalis*.

### 2.4. Disease Markers and Statistical Methods

#### 2.4.1. Statistical Analysis and Calculation of Disease Incidence, Disease Severity Index, and Lesion Area at Each Inoculation Site

In the experiment, each *P. alba* var. *pyramidalis* with spacings of 0.8 m × 0.8 m and 1.6 m × 1.6 m was inoculated at 30 sites, with 15 sites on both the top and the bottom of the tree. Each inoculation method was applied to three trees, totaling 90 inoculation sites. Disease incidence rate was calculated at 10, 20, 30, 40, and 50 days post inoculation. The disease incidence rate was calculated using Formula (1) [2]: Disease incidence rate (IR) = Number of the diseased inoculation sites/Total Number of inoculation sites × 100%. (1)

At every observation, each lesion’s longitudinal extension length (a, mm) and horizontal length (b, mm) were measured. Because the lesion shapes were mainly subelliptic, the lesion area of diseased the inoculation sites was calculated according to the elliptic area using Formula (2) [20]: (2)(A=a b π/4).

Additionally, due to variations in the healing area at control inoculation sites in the punching, burning, and toothpick inoculation experiments (0.60 ± 0.074 cm^2^, 0.85 ± 0.14 cm^2^, and 0.31 ± 0.088 cm^2^, respectively), the relative area of the lesions was adopted as a quantity standard, which is calculated using Formula (3) [3]: Relative area of lesion (*S_relative lesion_*) = Lesion area in the treatment group/healing area of inoculation sites in the control group.(3)

The symptoms of lesions were taken into consideration to determine the presence or absence of disease, and the disease severity index was calculated using Formula (4) [2]:Disease severity index (DI) = 100 × ∑ (Number of lesions at each grade × Specific grade value)/(Total number of inoculation sites × Highest grade value)(4)

The criteria for disease severity classification grading are presented as follows (Table 1):

#### 2.4.2. Observation and Measurement of Lesion Morphology and Spore Microstructure at Inoculation Sites

At 10, 20, 30, 40, and 50 days post inoculation, photographs were taken using a Canon 200D camera to document the lesion status and the quantity of pycnidia at each inoculation site. As black pycnidia grew on the lesions during inoculation, part of the structure was carefully collected using an inoculation needle and placed on a rice-paper plant, which was vertically dissected with a scalpel. Subsequently, the pycnidial structure was observed and photographed under an optical microscope (OLYMPUS BX51TRF, Olympus Corporation, Tokyo, Japan).

#### 2.4.3. Observation and Statistical Analysis of Wounded Area of Xylem at Inoculation Sites

At 50 days post inoculation, the rotten layer of the affected bark was removed using a scalpel to expose the xylem (white). Photographs were taken using a camera to observe the transversal section of the lesion. The length of both the longitudinal and transversal extensions of each lesion was measured with a ruler. Subsequently, a vertical incision was made perpendicular to the lesion surface, and photographs were taken to observe the longitudinal section of the lesion. The depth of the lesion-induced wound on the xylem was measured using a ruler. As the shape of the wounded xylem resembled an ellipse, the calculation for the wounded area of xylem followed the ellipse area formula (see Formula (2) in Section 2.4.1).

### 2.5. Data Processing and Statistical Analysis

In this study, the *t*-test was used to test the significance of the necrotic area (the relative area of lesion and xylem damage area on average) between burning inoculation and toothpick inoculation; ANOVA (analysis of variance) methods were used to test the significance of the relative lesion area under different intra-and inter-row spacing and inoculation positions between burning inoculation and toothpick inoculation; and the chi-square method was used to test the significance of the disease incidence rate between burning inoculation and toothpick inoculation.

## 3. Results and Analysis

### 3.1. Comparative Analysis of Symptoms of Lesions at Inoculation Sites Wounded by Punching, Burning, and Toothpick Inoculation Techniques

At 10 days of punching inoculation, watery changes were observed at the sites inoculated with *V. sordida*, and the surrounding periderm turned brown (Figure 1B). At days 20 to 30 post inoculation, circle-shaped necrosis exhibiting slow expansion was observed on the periderm, while gradual callus formation was visible in the xylem around the wounded holes (Figure 1C). At 40 days post inoculation, except for the longitudinal cracks at the center of the wound, the 5 mm inoculation hole was almost entirely filled with callus (Figure 1D). At 50 days post inoculation, the anatomical structure of the lesions showed the formation of secondary vascular bundles (xylem and phloem), creating a ring of brown necrotic tissue at the inoculation hole, penetrating deep into the original xylem surface. In contrast, signs of necrosis were observed in the controls. Additionally, irregularly distributed brown tissue was visible in the newly formed phloem and xylem, with uneven textures compared to the white, well-organized calluses formed in the controls. These results indicate that punching inoculation can induce symptoms of lesions in six-year-old *P. alba* var. *pyramidalis* in the early stages of *V. sordida* inoculation. With the development of disease resistance, the inoculation site is filled with newly formed callus, showing the capability of spontaneous healing, leaving a ring of necrotic peridermal tissue and partially necrotic xylem (Figure 1E).

At 10 days of burning inoculation, watery spots were observed at the inoculation sites, and the bark around the inoculation holes decayed and sank with color changes (Figure 1G). At 20 days post inoculation, the lesion enlarged with the appearance of longitudinal cracks. At 40 days post inoculation, the inoculation sites decayed, exhibiting symptoms such as water loss and shrinking, leading to a cracked appearance (Figure 1I). At 50 days post inoculation, the anatomical structure of the lesions revealed that the pathogen-induced necrosis extended approximately 1.2 cm into the xylem (Figure 1J).

At 10 days of toothpick inoculation, the periderm at the inoculation sites turned brown, softened, and sank (Figure 1L). At 20 days post inoculation, the decayed brown area around each inoculation site expanded, and the lesion extended longitudinally. Some lesions even began to produce numerous needle-like black protrusions, which were subsequently identified to be pycnidia of the pathogen (Figure 1M). At 40 days post inoculation, the inoculation sites showed severe symptoms, with the top and the bottom merging and most inoculation sites releasing rusty liquid (Figure 1N). At 50 days post inoculation, the anatomical structure of the lesions revealed that the pathogen caused severe damage to the xylem, and the wounded area was much larger than that in the burning inoculation group (Figure 1O).

In short, all three inoculation methods can induce symptoms of lesions in *P. alba* var. *pyramidalis* inoculated with *V. sordida*. However, there are differences in lesion status when comparing the punching inoculation method with the burning and toothpick inoculation methods. Punching inoculation resulted in early symptoms and spontaneous healing of the inoculation sites, whereas the burning and toothpick techniques both induced persistent and severe symptoms.

### 3.2. Phenomenon of Xylem Damage and Statistical Analysis of Xylem Damage Area at Inoculation Sites in Three Inoculation Methods

The wound on the xylem is shown in Figure 2, Figure 3 and Figure 4. Upon transversal sectioning of the lesions in the punching inoculation group, the xylem was minimally wounded. However, during dissection, the phloem and xylem displayed a rough appearance, with hard and coarse textures, distinct from the white, well-organized sections in the control group. Additionally, light brown spots were found on the treated section, indicating the potential presence of fungal hyphae, awaiting suitable conditions for infection (Figure 2A–C). The dissection at the longitudinal section of the lesions in the punching inoculation group revealed that, compared to the callus and newly formed green tissue on the wound surface of the controls (Figure 2D), the treatment group exhibited brown streaks extending deep into the xylem (Figure 2E). The observed callus and green tissue, in addition to symptoms at the inoculation sites in the early stages, indicate that, despite spontaneous healing, the punching technique, to some extent, causes wounding to the xylem. The most severe wound on the xylem was observed in the toothpick inoculation group (Figure 3). While the burning inoculation group also caused damage to the xylem to some extent at the inoculation sites, the wound was less severe than in the toothpick inoculation group (Figure 4).

Considering the potential impacts of intra- and inter-row spacing as well as inoculation positions on inoculation outcomes, the analysis of the area of wounded xylem (Figure 5A,B) was presented from the perspective of different inoculation positions (upper vs. bottom) and intra- and inter-row spacing (high vs. low), respectively. In the top of the poplar with high intra- and inter-row spacing (Figure 5A (left)) and the bottom of the poplar with low intra- and inter-row spacing (Figure 5B (right)), there was no significant difference in the area of wounded xylem between the burning and toothpick inoculation groups. In contrast, in the bottom of the poplar with low intra- and inter-row spacing (Figure 5A (right)) and the top of the poplar with high intra- and inter-row spacing (Figure 5B (left)), there was a significant difference in the area of wounded xylem between the burning and toothpick inoculation groups, with toothpick inoculation causing significantly larger area of wounded xylem than burning inoculation (*t*-test, *p* < 0.05).

In short, the toothpick and burning inoculation groups showed consistent patterns in the changes in the area of wounded xylem and symptoms of lesions at the inoculation sites. Toothpick inoculation of *V. sordida* induced severe damage to the xylem compared to burning inoculation.

### 3.3. Comparative Analysis of Disease Incidence, Disease Severity Index, and Relative Area of Lesion at the Inoculation Sites in the Burning and Toothpick Inoculation Groups

The toothpick and burning inoculation groups showed a gradual increase in the incidence of *Valsa* canker, both surpassing 50% at 20 days post inoculation. At 50 days post inoculation, lesions were observed at almost all inoculation sites (93.55%) in the toothpick inoculation group, and the incidence rate was significantly higher than 80.65% in the burning inoculation group (*p* = 0.007, *n* = 90, Figure 6A). The disease severity index exhibited the fastest increase in the toothpick inoculation group during the first 10 days, and the fastest at 20 days of inoculation in the burning inoculation group (Figure 6B). Changes in the relative area of the lesion of different inoculation groups at different time points are shown in Figure 6C. In the toothpick inoculation group, lesions were observed soon after inoculation, and the average lesion area was significantly greater than that in the burning inoculation group at 10 days post inoculation (*t*-test, *p* < 0.05). Subsequently, the lesion area at the inoculation sites gradually expanded over time. Between 30 and 40 days post inoculation, the relative area of lesion in the toothpick inoculation group further expanded, reaching a maximum value of 31.724. In contrast, the lesion area in the burning inoculation group remained stable after 30 days, with little further expansion in the later stages. At 50 days post inoculation, the mean relative area of the lesion was 7.02 ± 0.57 for the toothpick inoculation group and 1.86 ± 0.059 for the burning inoculation group.

To summarize, the disease incidence in the toothpick inoculation group was significantly higher than that in the burning inoculation group. Toothpick inoculation contributed to the fastest expansion of lesions and the largest relative area of lesions compared to the burning technique. On the other hand, burning inoculation displayed a lower disease severity index, and the relative lesion area was less than one-third of that induced by toothpick inoculation.

### 3.4. Effects of Intra- and Inter-Row Spacing as Well as Inoculation Positions on the Outcomes of Toothpick Inoculation

At 10 days post inoculation, the relative area of the lesions in the low intra- and inter-row spacing group was significantly higher than that in the high intra- and inter-row spacing group (ANOVA, *p* = 0.000009). The significant difference in the relative area of lesions between the high and low intra- and inter-row spacing groups further expanded from 20 to 50 days post inoculation (ANOVA, *p* ≤ 0.0001, Figure 7A). At 10 days post inoculation, there was no significant difference in the relative area of lesions between the top and the bottom of the wounded poplar. However, between 20 and 50 days post inoculation, the relative area of lesions in the top was significantly lower than that in the bottom (ANOVA, *p* < 0.05, Figure 7B).

In short, after toothpick inoculation of *V. sordida*, the poplars with low intra- and inter-row spacing were more susceptible to lesions compared to those with high intra- and inter-row spacing. Additionally, the bottom of the poplar appeared to be more susceptible to lesions compared to the top. These findings align with the results of the previous analysis on the impact of different stand densities and inoculation positions on the area of wounded xylem.

## 4. Conclusions and Discussion

### 4.1. Discussion on the Induction of Lesion by Punching, Burning, and Toothpick Inoculation of V. sordida

This study revealed that after the inoculation of *P. alba* var. *pyramidalis* trunk with *V. sordida* using the punching, burning, and toothpick techniques, *Valsa* canker symptoms were induced, and notable differences were observed in the induction of lesions between punching inoculation and the latter two inoculation methods. Previous studies have identified punching inoculation as a stable and efficient technique, making it an ideal option for molecular and physiological research [6]. However, in this study, punching inoculation showed less favorable inoculation outcomes. Moreover, increased wound healing and growth were observed after punching inoculation. This discrepancy may be explained by the use of six-year-old *P. alba* var. *Pyramidalis* with relatively high disease resistance in the current study. Calluses were found on the surrounding tissue of the lesions in the wounded poplars, preventing further expansion of the lesions. In addition, punching inoculation only wounded the periderm, with almost no impact on the underlying tissues. As a result, no necrotic tissue formed, resulting in a limited supply of saprotrophic nutrients to the pathogen [5]. In short, punching inoculation can produce average inoculation outcomes and induce callus formation. Compared to punching, the burning technique exhibited a stronger capability in inducing *Valsa* canker, with the damage caused by burning expanding from the center of the wound surface towards the surrounding and underlying tissues. This gradual extension from dead tissues to dying tissues, injured tissues, and healthy tissues is favorable for the pathogen. However, burning inoculation also presents certain challenges. To be specific, the duration of heating has a significant impact on lesions, with a high risk of producing errors, highlighting the importance of strict control over the duration of heating in burning inoculation [6]. Toothpick inoculation showed the highest disease incidence and disease severity index, reaching around 50% within 10 days after inoculation. In the later stages of the disease, it caused severe wounds to the phloem and xylem of *P. alba* var. *pyramidalis*. In this context, the remarkable inoculation outcomes may be attributed to the high vitality of the pathogen colonized onto the toothpicks [21]. Additionally, the depth of the wound caused by toothpick inoculation creates a relatively deeper saprotrophic environment than the other two inoculation methods, facilitating the invasion of *V. sordida* [5].

In the toothpick inoculation experiment, conspicuous sporulation was observed around the lesions. At 20 days post inoculation, numerous needle-like black protrusions (pycnidia at a density of 5.08 ± 1.33 per cm^2^) emerged on the lesions. At 30 days post inoculation, tangerine gelatinous filamentous structures were extruded from the pores of the pycnidia. At 50 days post inoculation, these structures gradually turned dark yellow and eventually disappeared (Figure 8A–D). During toothpick inoculation, the black protrusions were carefully collected using an inoculation needle. After longitudinal dissection, observation of the protrusions under a microscope revealed a large number of mature, elongated to sausage-shaped motile conidia within the black protrusions (Figure 8E). This finding suggests that toothpick inoculation can promote the reproduction of pathogenic spores. However, further analysis is needed to confirm whether this efficient induction of spore formation is associated with the physiological characteristics of the pathogen. In future studies, less-sporulative pathogens (e.g., *B. dothidea*) can be considered to validate the results in our study.

### 4.2. Effects of Different Stand Densities and Inoculation Positions on Inoculation Outcomes

The outcome of plant disease development depends on the interaction between the abiotic environment, the host, and the pathogen, namely the disease triangle [22]. In the abiotic environment, stand density, an agricultural factor determined by intra- and inter-row spacing, is associated with pathogen invasion into the host [23,24,25,26]. This study found that the intra- and inter-row spacing of *P. alba* var. *pyramidalis* can significantly affect the inoculation outcomes of *V. sordida*. Specifically, the disease severity index of the wounded *P. alba* var. *pyramidalis* with low intra- and inter-row spacing was significantly higher than that of the high intra- and inter-row spacing group, and the area of wounded xylem in the low intra- and inter-row spacing group was also significantly greater than that in the high intra- and inter-row spacing group. These results may be attributed to the low-level sunlight exposure and poor ventilation in the low intra- and inter-row spacing environment, and the resulting limited sunlight exposure for understory vegetation, leading to a low species diversity, decreased physiological functions of plants, weakened host resistance, and increased susceptibility to pathogen invasion [27]. Evidence from several studies indicates that shading increases infection by a range of pathogens [28]. Further research is needed to elucidate the relationship of disease development with sunlight exposure, crown density, and humidity.

For tall trees and other large plants, even the same organ may exhibit varying degrees of maturity due to its vertical distribution, leading to different levels of resistance to pathogens. Therefore, it is believed that different inoculation positions can, to some extent, affect the susceptibility to pathogens. In this study, the outcomes of toothpick inoculation varied significantly in different inoculation positions. After toothpick inoculation, the disease severity index of the bottom was markedly higher than that of the top, and the area of wounded xylem in the bottom was also greater than in the top. This may be explained by the uniform diameter of the inoculated trees in both top and the bottom, and the bottom, being closer to the ground, is affected by the understory vegetation or litter cover, thus exhibiting higher susceptibility to the disease compared to the top [29,30].

Additionally, tree age is another key factor in disease development [31,32]. Reportedly, *Valsa* canker is more likely to occur and poses a greater threat in young poplar forests [33]. Therefore, an additional burning inoculation experiment was conducted using two-year-old *P. alba* var. *pyramidalis*, with the discovery of lesions induced by both CZC and CZA, while no sinking of the wound or expansion of the inoculation holes was observed in the control group. Notably, callus formation was visible during the inoculation process (Figure 9A). The treatment groups exhibited longitudinal sinking of lesions and separation of fibrous tissue in the lesions during later stages. The CZC-induced lesion was more severe compared to the CZA-induced lesion (Figure 9B–E). The incidence of the CZC-induced lesion reached 100%, and the disease severity index was 82.81, significantly higher than the incidence of *Valsa* canker (80.65%) and the corresponding disease severity index (35.48) in the six-year-old *P. alba* var. *pyramidalis* with burning inoculation (Table 2). Collectively, the development of lesions induced by burning inoculation in two-year-old *P. alba* var. *pyramidalis* is faster and more severe than that in six-year-old *P. alba* var. *pyramidalis*, possibly due to the weaker resistance of young seedlings. These findings are consistent with previous reports [34,35] suggesting a higher degree of disease severity index in the case of burning inoculation using poplars of a younger age.

### 4.3. Application of Toothpick Inoculation

Toothpick inoculation is an efficient technique that allows for the rapid expansion of lesions. Analysis can be conducted at 10 days post inoculation, making it an easy-to-use, convenient method for large-scale inoculation experiments in the study of diseases in branches and trunks. Despite its advantages, there are limited applications of toothpick inoculation in trunk disease research. Ghimire et al. (2019) used toothpick inoculation and trunk wounding techniques to inoculate soybeans with *Diaporthe* spp. hyphae, and found that the length of lesions and the severity of disease induced by toothpick inoculation were significantly higher than those induced by trunk wounding, which were consistent with the findings of this experiment [9]. Yue et al. (2011) also investigated the resistance of 57 species and varieties (clones) of poplars, and *P. alba* var. *pyramidalis* was identified as a species with strong resistance [36]. In the current study, toothpick inoculation yielded the optimal results, indicating applicability to most inoculation experiments using poplars. However, toothpick inoculation, considering the length of toothpicks, is only suitable for inoculating plants with a diameter of at least 2 cm. Additionally, toothpick inoculation can be used for rapid lesion induction in the field. This can facilitate the study of the growth, development, spore production, infection, nutritional metabolism, or pathogenic processes of a pathogen in the natural environment, thereby boosting the advancement of pathological research on diseases in branches and trunks.

## Figures and Tables

**Figure 1 biology-13-00251-f001:**
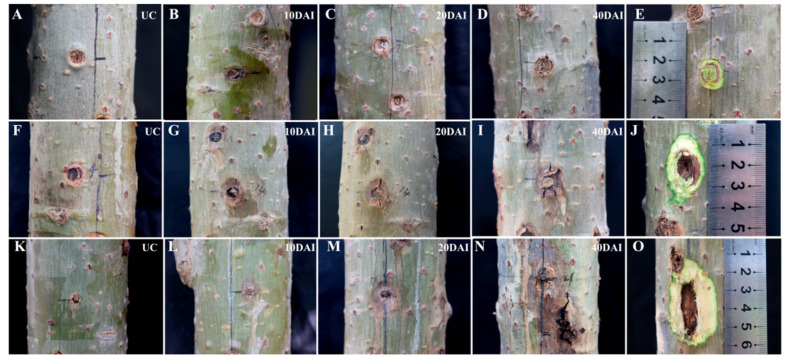
Cross-sectional diagrams of lesion formation after punching, burning, and toothpick inoculation. Note: (**A**–**E**) Punching inoculation of *V. sordida*; (**F**–**J**) burning inoculation of *V. sordida*; (**K**–**O**) toothpick inoculation of *V. sordida*. (**A**,**F**,**K**) Controls inoculated with water–agar blocks; (**B**–**D**,**G**–**I**) and (**L**–**N**) development of lesions at 10, 20, and 40 days post inoculation; (**E**,**J**,**O**) cross-sectional diagrams of lesions at 50 days post inoculation with punching, burning, and toothpick techniques, respectively.

**Figure 2 biology-13-00251-f002:**
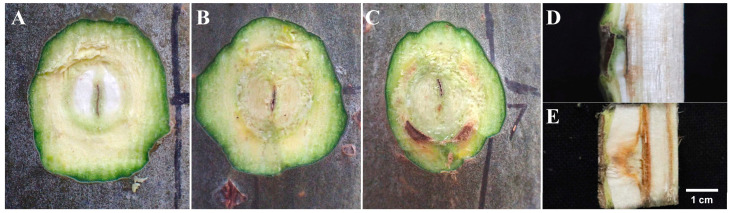
Cross-sectional diagrams of the inoculation sites after punching inoculation at 50 days post inoculation. Note: (**A**) cross-sectional diagram of the inoculation sites in the control group; (**B**,**C**) transversal cross-sectional diagrams of the lesions in the treatment group; (**D**,**E**) longitudinal cross-sectional diagrams of the lesions in the treatment group.

**Figure 3 biology-13-00251-f003:**
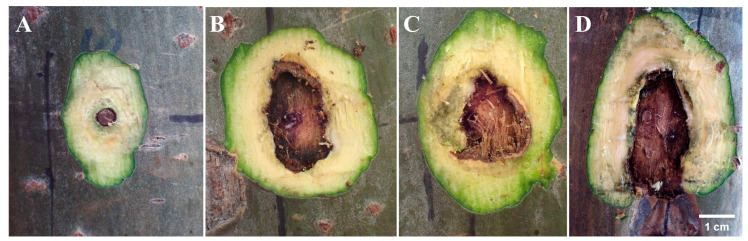
Cross-sectional diagrams of inoculation sites with toothpick inoculation at 50 days post inoculation. Note: (**A**) cross-sectional diagram of the inoculation sites in the control group; (**B**–**D**) transversal cross-sectional diagrams of the lesions in the treatment group.

**Figure 4 biology-13-00251-f004:**
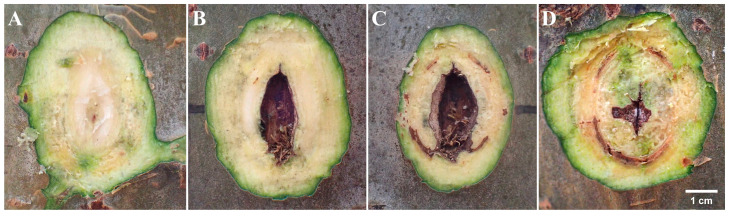
Cross-sectional diagrams of inoculation sites with burning inoculation at 50 days post inoculation. Note: (**A**) cross-sectional diagram of the inoculation sites in the control group; (**B**–**D**) transversal cross-sectional diagrams of the lesions in the treatment group.

**Figure 5 biology-13-00251-f005:**
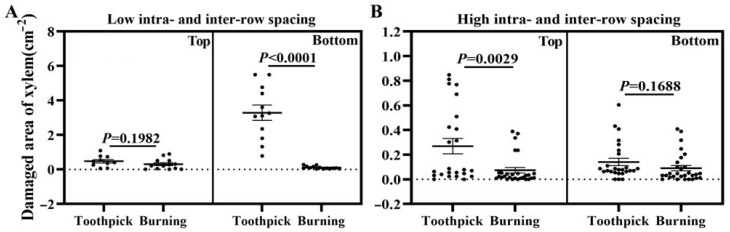
Area of the wounded xylem at the inoculation sites after toothpick and burning inoculations. Note: (**A**) inoculation on *P. alba* var. *pyramidalis* with low intra- and inter-row spacing (*n* = 15 inoculation sites); (**B**) inoculation on *P. alba* var. *pyramidalis* with high intra- and inter-row spacing (*n* = 30 inoculation sites); top (**left**) and bottom (**right**); Top, inoculation in the top of the poplar; Bottom, inoculation in the bottom of the poplar; bars indicate mean ± SEM, *p* values are shown, and *p* < 0.05 indicates significant differences between the treatments; The black transverse line in bars represent the mean value and the black vertical line in bars represent the SEM value; The black lines below the *p* values represent the comparison of significance between the burning and toothpick inoculation groups.

**Figure 6 biology-13-00251-f006:**
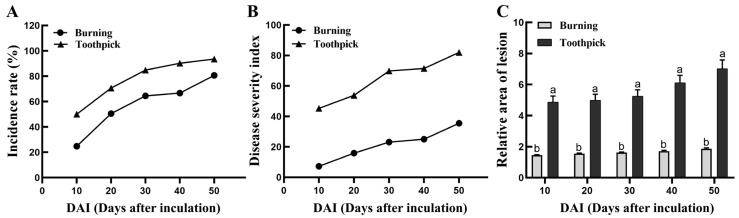
Changes in disease incidence, disease severity index, and relative area of lesion at the inoculation sites in the burning and toothpick inoculation groups. Note: (**A**) the disease incidence rate of the burning and toothpick inoculation groups at different time points. (**B**) the disease severity index of the burning and toothpick inoculation groups at different time points. (**C**) the average relative area of lesions in the burning and toothpick inoculation groups at different time points. Bars indicate mean ± SEM and different letters indicate significant (*p* < 0.05, *n* = 90 inoculation sites).

**Figure 7 biology-13-00251-f007:**
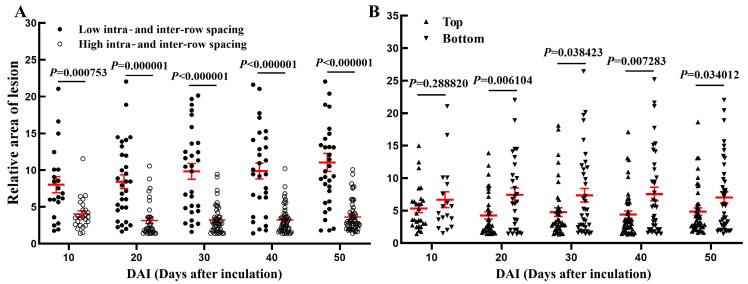
Summary of relative areas of lesions at different stand densities and inoculation positions. Notes: (**A**) Low intra- and inter-row spacing: inoculation in the poplar of Low intra- and inter-row spacing (*n* = 30 inoculation sites); High intra- and inter-row spacing: inoculation in the poplar of high intra- and inter-row spacing (*n* = 60 inoculation sites); (**B**) Top: inoculation in the top of the poplar (*n* = 45 inoculation sites); Bottom: inoculation in the bottom of the poplar (*n* = 45 biologically independent samples); bars indicate mean ± SEM, *p* values are shown, and *p* < 0.05 indicates significant differences between the treatments. The red transverse lines in bars represent the mean value and the red vertical lines represent the SEM value; The black lines below the *p* values represent the comparison of significance between the burning and toothpick inoculation groups.

**Figure 8 biology-13-00251-f008:**
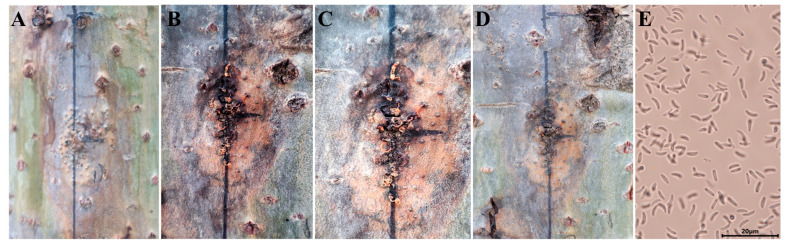
Process of changes in the production of pycnidia induced by toothpick inoculation. Note: (**A**–**D**) production of pycnidia by the lesions; (**E**) conidia under microscope.

**Figure 9 biology-13-00251-f009:**
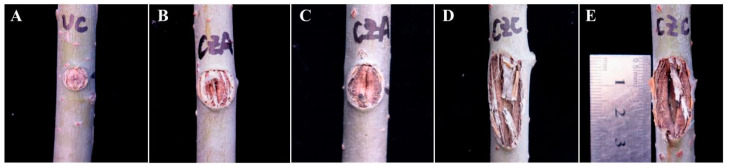
Changes in the inoculation sites on the trunks of two-year-old *P. alba* var. *pyramidalis* after burning inoculation with CZA and CZC. Note: (**A**) inoculation sites in the control group; (**B**,**C**) inoculation sites in the CZA-treated group; (**D**,**E**) inoculation sites in the CZC-treated group.

**Table 1 biology-13-00251-t001:** Classification of disease severity.

Grade	Lesion Area in the Treatment Group/Area of Inoculation Sites in the Control Group (*S_relative area_*)
0	No infection in the treatment group (*S_relative area_* ≤ 1.25)
1	The lesion area in the treatment group is 1.25 to 1.75 times that of the control inoculation sites (1.25 < *S_relative area_* ≤ 1.75)
2	The lesion area in the treatment group is 1.75 to 2.25 times that of control inoculation sites (1.75 < *S_relative area_* ≤ 2.25)
3	The lesion area in the treatment group is 2.25 to 2.75 times that of the control inoculation sites (2.25 < *S_relative area_* ≤ 2.75)
4	The lesion area in the treatment group is over 2.75 times that of the control inoculation sites (*S_relative area_* > 2.75)

**Table 2 biology-13-00251-t002:** Summary of CZA- and CZC-induced lesions in two-year-old *P. alba* var. *pyramidalis* after burning inoculation.

	Treatment	Incidence Rate, %	Disease Severity Index	Relative Lesion Area
1	CZA	18.75	30.08	1.65 ± 0.40
2	CZC	100	87.50	3.58 ± 1.38 ****

Note: **** indicates *p* < 0.00001, *t*-test.

## Data Availability

Data are contained within the article.

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
