# Peer review of "Comparative Study on the Effectiveness of Three Inoculation Methods for Valsa sordida in Populus alba var. pyramidalis"

_biology, 2024, doi:10.3390/biology13040251_

Round 1
Reviewer 1 Report
Comments and Suggestions for Authors
This is a potentially valuable and relevant study but needs considerable work to achieve the necessary clarity and soundness of methods required for publication. Below are comments and questions on the manuscript in order of appearance.
lines 54-56. Give common names and brief description of damage caused by the pathogens; is it in the foliage? stems? roots? Give this information here, early in the presentation.
l. 60 what is the "burning technique"? Although presented later, an early description of the techniques is better.
l. 128 describe a little more in detail the "5 mm punch" and depth of the hole made with it.
l. 134 is 5 cm. sufficient distance between inoculation site and control to prevent cross-contamination or infection of control tissue
l. 137 what is a "fungal clump" and how was it inoculated?
l. 162 how deep was the hole drilled and how does it compare with the depth of the punch holes?
l. 187 give formula, not "as follows" with a citation! same for l. 191 and 194.
l. 191-194. Not at all clear how control lesions were used to compare to inoculated lesions to produce data on disease incidence.
l. 195 what are "symptoms of lesions"? how did you distinguish between infected lesion and mechanical lesion? Is it only a function of lesion area as given in Table 1? Pycnidia are only mentioned later in line 207, but not whether or how they were utilized.
lines 198-203 definitions of lesion area and disease severity are not very clear. For example, if lesions are roughly circular why isn't the formula for the area of a circle used? If not circular, what is the rationale for Formula 2 in l. 198? Clarify. Clarify how how disease severity is being quantified - is it a % measure? Or just an index? Graphs presented in results have unclear units on the Y axis.
l. 222 chi square for what data? No frequency data amenable to chi square have been described. Similarly, ANOVA for which factor and which factor levels? Specifiy the experimental design and response variables used. As stated, the description of statistical methods is meaningless and the presentation of statistical results later in the manuscript is totally unclear.
l. 227 mean +/- std. deviation is NOT THE SAME AS mean +/-std. error of the mean!! The latter should be reported and graphed.
Figure 1. Not having pictures of the controls over time (10, 20 and 40 days) as with inoculations makes inoculated sites impossible to compare and interpret vs. controls
l. 278 Sub-section 3.2 heading on statistical analysis is followed immediately by Figure 2 - bad placement
Figure 5 has several problems: Y axis gives units as cm -3 which makes no sense; P values with no clearly defined and associated statistical test are not sufficient for judging significance; labels of "top" or "bottom" and caption make no sense.
Figure 6. Panel C has letters a and b with no explanation or reference to a statistical test. As mentioned earlier, the methods sections does not clearly explain experimental design or statistical testing.
l. 350. relative area of lesion reached values of 31.724 but Y axis in Fig. 6 C goes from 0-10. ???? Unfortunately, this very careless and unclear presentation of results is a common flaw of this whole manuscript.
Figure 7. impossible to interpret in view of the same problems mentioned for previous figures.
l. 488 how valid is it to extrapolate to the other diseases based on this study?
Comments on the Quality of English Language
Mostly ok.
Reviewer 2 Report
Comments and Suggestions for Authors
In the discussion of results the authors should consider that, the original size of the wound is smaller in the punching, than in that made with toothpick, therefore, this may favor more the development of the infection
In materials and methods the observations are noted in the attached manuscript

Round 2
Reviewer 1 Report
Comments and Suggestions for Authors
Authors have made some significant improvements, but a few of the key problems that I pointed out remain in the new version.
Section 2.5. The explanation of statistical testing is still not clear. It shouldn't be necessary to explain this, but I will: A t-test compares two levels of a given factor to determine if they differ in their effect on the response variable. Here the authors seem to refer to the response variable (necrotic area) but don't specify the two factor levels being used. The ANOVA explanation is better worded, but the the use of chi-square remains unclear. Chi-square is used to test whether the frequency of the observations differ among categories of a factor. Is the frequency of the disease incidence rate the response being tested here? What are the categories being compared?
Figure 5. The authors should give a consistent measure of variability for means, and the preferred measure is the SEM. The mix of SD and SEM is bad presentation and inconsistent.
Figure 6 panel C. While authors clarify in their comments that they are referring to the average area of lesion, this needs to be specified not in the text of the paper but importantly in the figure legend.
These corrections and clarifications are not difficult but do need to be made.
Comments on the Quality of English Language
OK
Round 3
Reviewer 1 Report
Comments and Suggestions for Authors
The manuscript is now acceptable given the latest round of corrections.
Comments on the Quality of English LanguageMinor copy editing may be required.